# Impact of Seismic Activity on Access to Health Care in Hispanic/Latino Cancer Patients from Puerto Rico

**DOI:** 10.3390/ijerph19074246

**Published:** 2022-04-02

**Authors:** Cristina Peña-Vargas, Yoamy Toro-Morales, Paola Valentin, María López, Zindie Rodriguez-Castro, Ruthmarie Hernandez-Torres, Nelmit Tollinchi-Natali, Normarie Torres-Blasco, Cristina Pereira, Guillermo N. Armaiz-Pena, Heather Jim, Eida M. Castro-Figueroa

**Affiliations:** 1Ponce’s Research Institute, Ponce Health Sciences University, Ponce 00716, Puerto Rico; zrodriguez@psm.edu (Z.R.-C.); normarietorres@psm.edu (N.T.-B.); garmaiz@psm.edu (G.N.A.-P.); ecastro@psm.edu (E.M.C.-F.); 2School of Behavioral and Brain Sciences, Ponce Health Sciences University, Ponce 00716, Puerto Rico; ytoro19@stu.psm.edu (Y.T.-M.); pvalentin19@stu.psm.edu (P.V.); ntollinchi@psm.edu (N.T.-N.); cpereira14@stu.psm.edu (C.P.); 3School of Medicine, Ponce Health Sciences University, Ponce 00716, Puerto Rico; mlopez20@stu.psm.edu; 4Clinical and Translational Science Institute, University of Rochester, Rochester, NY 14642, USA; ruthmarie_hernandez@urmc.rochester.edu; 5Moffitt Cancer Center, Department of Health Outcomes and Behavior, Tampa, FL 33612, USA; heather.Jim@moffitt.org

**Keywords:** cancer, barriers to health care, natural disaster, psycho-oncology, health disparities, access to health care

## Abstract

On 7 January 2020, the southern region of Puerto Rico was struck by a 6.4 magnitude earthquake, followed by continual seismic activity. Our team performed secondary analyses to explore the relationship between exposure to seismic activity, protection (support) received, and barriers to health care access for cancer patients. Methods: The research team collected data from the database of a longitudinal case-control cohort parent study concerning the impact of Hurricane Maria in Puerto Rican cancer patients. The participants from the parent study were recruited in community clinics. The extracted data was collected from 51 cancer patients who completed the parent study’s interviews from January–July 2020 (seismic activity period). Barriers to health care were assessed using the Barrier to Care Questionaries (BCQ), which is composed of five subscales: skills, marginalization, knowledge and beliefs expectations, and pragmatics. Exposure to seismic activity and protection was assessed using their respective subscales from the Scale of Psychosocial Impact of Disasters. Results: The results showed a significant relationship between exposure to seismic activity and barriers to health care (*p* < 0.001) and its five subscales (*p* < 0.01). These results shed light on potential access to care barriers that could hinder cancer patient treatment in the event of a natural disaster.

## 1. Introduction

On 7 January 2020, the southern region of Puerto Rico was impacted by a 6.4 magnitude earthquake [1], followed by a continual seismic activity that caused structural damage to buildings, houses, and roads [2]. Puerto Rico is located near two tectonic plates which make the island susceptible to experiencing seismic activity. However, more than a hundred years had passed since the island was struck by an earthquake of this magnitude [3]. It is important to highlight that even before the seismic activity of 2020, the island was suffering from infrastructure deficiencies associated with the aftermath of Hurricane Maria (landfall on 20 September 2017). Moreover, an evaluation carried out by the American Society of Civil Engineers in 2019 rated Puerto Rico’s infrastructure as poor, exhibiting significant deterioration, and at high risk of failure [4]. Natural disasters can have a devastating impact on access to medical services [5], causing infrastructural damage (damage to roads, power system, water supply, buildings, among others) shown to exacerbate symptoms associated with chronic diseases and have a long-lasting negative impact on patients’ health [6,7].

Likewise, it has been reported that barriers to health care such as lack of access to hospitals, cancer centers, infrastructure damage, and limited communication are widespread in the aftermath of a natural disaster [8]. These barriers increase the risk for treatment interruption in patients with chronic health conditions, especially cancer patients [9], as they require extended and multimodal regimens. For example, in the aftermath of Hurricane Katrina in 2005, cancer patients reported difficulty obtaining care, as they faced limited transportation options [10]. Disruption in treatment could lead to increased symptom burden and disease progression [11,12]. Another study reported the impact of Hurricane Maria on Puerto Rico’s health care system and revealed that at one point in time, 89% of health care sites were operating under challenging conditions, and only 13% of 70 assessed sites had electrical power [13]. In addition, damages to the infrastructure (transportation services and damaged roads, among others) persisted up to six months after the hurricane [14]. As the impact of natural disasters has been previously observed on the island, it is important to consider how treatment interruption and prolongation can worsen the condition and the outcomes of patients, which could result in poorer survival rates.

Scientific literature is scarce regarding the psychosocial and health care impact of natural disasters among cancer patients, particularly Hispanic/Latinos. The 2020 earthquake and aftershocks caused structural damage to buildings and homes in the southern region of Puerto Rico, including health care facilities, potentially exacerbating barriers to access and quality of health care. Exploring such an impact could help develop target protocols and procedures to address the health care needs of cancer patients after a natural disaster, which are an undeniable risk as cancer is the second leading cause of death amongst Puerto Ricans, with a 157.1 per 100,000 mortality rate [15]. The research team conducted secondary analyses to evaluate if the exposure to danger during the seismic activity predicted barriers in the access to health care services, and to explore if the support received during and after the seismic activity predicted fewer barriers to access to health care.

## 2. Materials and Methods

To test the proposed variables, the team conducted secondary analyses derived from a longitudinal case-control cohort study. The parent study (2U54MD007579-34) aims to describe unmet needs among Puerto Rican cancer patients exposed to Hurricane Maria, exploring post-Maria multilevel barriers and facilitators of access to cancer care, and assessing physiological (e.g., circulating markers of stress, hair cortisol, and catecholamines) and psychological markers of stress (e.g., anxiety, depression, post-traumatic stress disorder) linked to the hurricane. Participants were recruited in collaboration with community partners that included local community oncology clinics, cancer support groups, the Puerto Rico American Cancer Society, and community-based cancer support service organizations. The study divided participants into two groups: controls and cancer patients. For the control group, inclusion criteria were to be (1) 21 years or older, (2) capable of reading and speaking Spanish, and (3) able to provide informed consent. The exclusion criteria for this group were to not have a current diagnosis of cancer or history. For the cancer patient group, the same inclusion criteria were established with the addition that the participants had to be under the care of a physician for their cancer during the post-Maria critical period (within the first six months after the hurricane). Follow-up assessments were conducted every three months, for up to a year.

The research team collected data from the database of the parent study. The extracted data were collected from 51 cancer patients who had completed the Barriers to Care Questionnaire and the Scale of Psychosocial Impact of Disasters, SPSI-D, from January to June 2020. This period was selected as the seismic activity intermittently prolonged for approximately six months. The Barriers to Care Questionnaire (BCQ) [16] was used to assess barriers confronted by patients in a multiservice community clinic. This scale was translated to the Spanish language by the team, and a reliability test was conducted. For the total scale, the reliability test showed a Cronbach’s alpha of 0.97, which suggests strong reliability. The Barriers to Care Questionnaire evaluates circumstances that may interfere with access or use of health care services. It comprises 39 items, divided into five subscales: pragmatics, skills, expectations, marginalization, and knowledge and beliefs. The pragmatics subscale focuses on accessibility, quality, and cost issues when acquiring medical help from physicians, which might prevent or delay services (e.g., “Getting to the doctor’s office”). The skills subscale centers on the strategies a patient acquires to navigate and function within the health care system (e.g., “Understanding doctor’s orders”). The expectations subscale focuses on the expectations regarding the quality of care received from physicians or the health care system (e.g., “Mistakes made by doctors or nurses”). The marginalization subscale centers on the negative experience within the health care system, especially during visits (e.g., “Getting the doctor to listen to you”). The knowledge and belief subscale focuses on the patient and physician’s different ideas and beliefs about the nature and treatment of an illness (e.g., “Disagreeing with the doctor’s orders”). The questionnaire incorporates a Likert scale in which higher scores indicate better access to care and lowers scores indicate poorer access.

To evaluate exposure to danger during the seismic activity period of January (2020), the Escala de Impacto Psicosocial de los Desastres (Scale of Psychosocial Impact of Disasters, SPSI-D) [17] was used. This inventory was developed and validated in Chile to measure disruptive and healthy responses after a natural disaster. It includes 51 items based on a Likert scale, grouped into six dimensions: exposure (exposición), protection (apoyo), optimistic or positivist beliefs (creencias de suerte), positive self-concept (auto-concepto positivo), disruption (disrupción), and health (salud). For these secondary analyses, only the exposure and the protection subscales were used, as the other subscales did not measure the variables in question. The exposure subscale measures the disaster impact regarding material loss, property damage, disaster-driven isolation, and disaster-related injuries. The protection subscale measures the support received, such as emotional support, spiritual support, and information and guidance about what to do after the disaster. Univariate statistical analysis was conducted to describe the sample’s sociodemographic and clinical characteristics. Linear regression analyses were conducted to determine predictors for the barriers in health care access. In addition, Spearman correlation analyses were run to evaluate the relationship between scores regarding earthquake exposure and protection, and the BCQ’s five subscales. Analyses were made using a 0.05 alpha level and were conducted using IBM-SPSS Version 28 statistical analysis software.

## 3. Results

### 3.1. Sociodemographic and Clinical Characteristics of the Sample

The results show that women have been more compliant when accepting to collaborate in research, as 82% of the pooled sample were women. The average age of the participants was $\bar{x}$ = 60. Less than a quarter of the participants were employed (18%). On the other hand, a significant portion of the sample reported that their income was equal to or below $19,000 a year (70%). Less than a third of participants indicated that the money influx entering their household was enough to cover their needs (26.5%). The education level of most of the sample was high school or higher (88%). Most of the participants had breast cancer (44.9%), followed by uterus (12.2%), endometrium (6.1%), thyroid (4.1%), intestines (4.1%), prostate (4.1%), skin (4.1%), and others (20%). A total of 21.6% of the sample had metastatic disease. Most participants had had surgery to remove the tumor (85.4%), and in terms of treatment, more than half were receiving or received chemotherapy (67.3%), followed by radiation therapy (33.3%) and hormone therapy (12.5%). Table 1 shows a detailed description of the sociodemographic data.

### 3.2. Relationship between Barriers to Health Care, Exposure, and Protection

A linear regression analysis showed a significant relationship between exposure to danger during the earthquakes and access to health care (*p* < 0.001), indicating that exposure to seismic activity was a predictor for barriers in the access to health care services. The adjusted correlation coefficient was 0.317, which indicates that exposure to seismic activity explains 31.7% of the variance for the barriers to health care variable. On the other hand, no significant relationship was found between protection and barriers to health care (*p* = 0.224), which suggests that the protection received during or after the seismic activity did not impact the barriers in the health care services received by the cancer patients. Bellow, Table 2 shows a detailed description of the regression models.

### 3.3. Relationship between Barriers to Care, Exposure, and Protection during and after the Seismic Activity

We assessed the five dimensions of the BCQ to determine which dimensions had the most relationship with exposition to seismic activity. A correlation analysis was conducted to assess the relationship between earthquake exposition and the barriers to care subscales. Results showed a significant negative correlation between the exposure to seismic activity and skills subscale r(50) = −0.624, *p* < 0.001, with a strong effect size. It also showed a significant and strong negative correlation between exposure and the Pragmatics subscale r(50) = −0.600, *p* < 0.001, the Knowledge and Beliefs subscale r(50) = −0.519, *p* < 0.01, and the Marginalization subscale r(50) = −0.410, *p* < 0.01. Lastly, results showed a significant and moderate negative correlation between exposure and the Expectations subscale r(50) = −0.370, *p* < 0.01. Moreover, no significant relationship was found between the protection subscale and the BCQ’s subscales. Table 3 shows a detailed description for the correlation analyses.

## 4. Discussion

Recent literature shows that one of the primary concerns of cancer patients following a natural disaster is access to treatment [18]. Moreover, a study conducted on cancer patients showed that Hurricane Maria caused long-lasting effects on barriers to health care [19]. This data supports the need for investment in healthcare infrastructure to ease access to health care after a natural disaster. Although during the seismic activity, access to health care was provided through the opening of ambulatory clinics at temporary refugee camps, such as medical and psychological services [20,21], the team considered it crucial to explore if the access to health care was hindered in other ways. The first goal of these secondary analyses was to assess the relationship between barriers to health care services and exposure to seismic activity throughout January–July 2020. The relationship between exposure to seismic activity and barriers to care was found to be significant according to the results of the regression analysis. The results suggest that exposure to seismic activity was related to higher barriers to care in this study’s sample.

The analysis performed to assess the correlation between earthquake exposure and each of the five subscales in the BCQ shed light on the nature of these barriers. The subscale with the strongest correlation was the Skills subscale, suggesting that study participants could have had difficulty with the technical vocabulary used by the providers and staff, trouble filling out forms, and lack knowledge regarding different health care protocols or procedures. Interestingly, this could highlight a fault in the system, suggesting that the health care system fails to make the information transmittable for the layperson. Research suggests that low health literacy is related to poor condition management [22]. Inadequate health literacy is also related to health status and clinical outcomes, such as a higher risk for comorbidities, need of assistance, and depression and anxiety [23]. Following the Skills subscales, the Pragmatics subscales also showed a strong correlation. This subscale assessed the logistical and cost issues that could cause delays in access to health services. After the seismic activity, some roads were blocked, and the power system was temporarily affected, which could explain logistical issues to access to health care.

Furthermore, the Knowledge and Beliefs subscale of the BCQ also had a significant positive correlation with earthquake exposure. This suggests that, while receiving care during the earthquake and aftershocks’ exposure period, participants had incorrect beliefs about the treatment or recommendations received from their providers or the staff. It is crucial for health care providers to ensure that their messages and recommendations are delivered, especially during stressful moments such as natural disasters. Adequate education of patients is beneficial for symptom reduction and quality of life [24]. Findings for the Marginalization subscale imply that participants exposed to the seismic activity had several negative experiences within the health care system regarding how the providers and staff treated them. This could suggest that the participants perceived apathy, neglect, and indisposition, amongst other types of mistreatments. These are significant findings as poor quality of service could affect the treatment course of cancer patients.

Lastly, the relationship between the BCQ’s Expectations subscale and the exposure to seismic activity suggests that patients were not satisfied with the health care services received during the seismic activity period, as it potentially did not meet their expectations. These results can also be associated with the quality of service. It is essential to further investigate the quality-of-service patients receive after being exposed to a natural disaster. Literature shows that logistics and pragmatics tend to have priority in disaster management and resolution [25]. Notwithstanding, the quality of service is just as crucial. Studies have shown that patient satisfaction is associated with higher survival rates [26], making the quality of service a possible protective factor amongst cancer patients. It would be ideal for further studies to assess the perceived quality of the services from the providers’ point of view after a disaster. Studies evaluating the potential multilevel barriers hindering the quality of health care services after a natural disaster are also warranted. For instance, one angle that could be explored is the stress posed on providing care during the aftermath of natural disasters and how this burden could potentially impact the quality of care and the patient–provider relationship. Lastly, no significant relationship was found between protection and the barriers to health care. These results highlight the need to further investigate protection and support after a natural disaster. It is crucial to explore which are the specifics needs of cancer patients after a natural disaster to offer adequate protection and support.

One of the limitations of this investigation was that potential covariates were not assessed as they were not accounted for in the parent study. Future research should be able to include an assessment of the impact of these potential covariates. Another limitation was that the BCQ has not been translated in Spanish and had to be translated by the team. Adapting the BCQ for the Puerto Rican population would be ideal for future studies. Another future recommendation is directly assessing the efficiency of mobile clinics. During the seismic activity aftermath, there were services available through mobile clinic units, and cancer patients were transferred to clinics and hospitals outside of the earthquake’s impact area, such as nearby municipalities that were less or unaffected. Likewise, community-based organizations (e.g., the American Cancer Society–Puerto Rico) and other community clinics from different parts of Puerto Rico responded by attending to basic care needs [27,28]. Notwithstanding these limitations reported, the study team decided to conduct these analyses, as they provide important information about the access to health care during a natural disaster that must be addressed. This assessment reveals a critical angle that has been understudied in Puerto Rico, as the potential impact of natural disasters on health care quality is a fundamental theme that should be explored to identify specific issues to resolve and improve for the benefit of the quality of life of cancer patients.

## 5. Conclusions

The results of the analyses indicated that there is a strong significant relationship between the levels of exposure to earthquake-related dangers with barriers of access to health care services. However, no significant relationship was found between barriers to care and protection. These results shed light on potential access to care barriers that could hinder cancer patient treatment in the event of a natural disaster. Most importantly, the results proved significant information regarding the relationship of different types of barriers regarding the topics of skills, knowledge and beliefs, marginalization, and expectations of cancer patients after a natural disaster-related crisis. This information is vital when addressing disparity gaps among cancer patients in Puerto Rico. The team believes this information can help improve how to manage cancer patients’ specific needs during and after a natural disaster to minimize the risk of symptom progression or exacerbation.

## Figures and Tables

**Table 1 ijerph-19-04246-t001:** Sociodemographic characteristics of the sample (*n* = 51).

Variables	Mean (SD)/*n* (%)
** *Socio-demographic characteristics* **	
Age	60.54 (12.37)
Sex (female)	41 (82%)
Marital status (married)	20 (40%)
Employment status (employed)	9 (18%)
Income (≤12,001–19,000)	35 (70%)
Income enough (yes)	13 (26.5%)
Education level (high school or higher)	44 (88%)
** *Cancer Diagnosis Clinical Characteristics* **	
Type of cancer	
Breast	22 (44.9%)
Uterus	6 (12.2%)
Endometrium	3 (6.1%)
Thyroid	2 (4.1%)
Intestines	2 (4.1%)
Prostate	2 (4.1%)
Skin	2 (4.1%)
Others	10 (20%)
Cancer treatment	
Surgery	41 (85.4%)
Chemotherapy	33 (67.3%)
Radiation therapy	16 (33.3%)
Hormone therapy	6 (12.5%)
Metastasis (yes)	8 (21.6%)

**Table 2 ijerph-19-04246-t002:** Regression analysis to determine the relationship between exposure to earthquakes and barriers to health care services (*n* = 51).

Models for Barriers to Health Care	R^2^	F	B	*p*
Exposure	0.331	24.23	−3.32	0.000
Protection	0.033	1.515	−0.820	0.224

**Table 3 ijerph-19-04246-t003:** Correlation analysis to evaluate the relationship between barriers to care, exposure, and protection (*n* = 51).

Variables	M	SD	Variables	
			Exposure	Protection
1. Exposure to seismic activity	2.55	2.78	-	-
2. Protection during and after seismic activity	3.53	3.39	-	-
3. Total barriers to health care	88.94	16.05	−0.575 **	−0.173
4. Skills subscale	91.98	16.25	−0.620 **	−0.140
5. Marginalization subscale	91.45	17.80	−0.410 **	−0.218
6. Expectation subscale	89.23	20.55	−0.370 **	−0.155
7. Knowledge and beliefs subscale	92.21	15.10	−0.519 **	−0.135
8. Pragmatics subscale	82.58	19.94	−0.600 **	−0.094

** *p* < 0.01.

## Data Availability

The data presented in this study are openly available in https://figshare.com/articles/dataset/DataBase_BCQ_SPSI_sav/19142039 (accessed on 16 March 2020).

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
