# Peer review of "Impact of Seismic Activity on Access to Health Care in Hispanic/Latino Cancer Patients from Puerto Rico"

_ijerph, 2022, doi:10.3390/ijerph19074246_

Round 1

Reviewer 1 Report

In the manuscript titled "Impact of seismic activity on access to health care in His-2 panic/Latino cancer patients from Puerto Rico", the authors have attempted to explore the relationship between earthquake related dangers and barriers to healthcare, by conducting assessment on cancer patients who had been exposed to the Hurricane Maria in Puerto Rico. 

Major concerns: 

  • Materials and Methods (line 77-78): “To test the proposed variables, the team conducted secondary analyses derived from a longitudinal case-control cohort study”. Please provide reference to the parent study, if the study has been published.
  • Materials and Methods: In the section describing the “Barriers to Care Questionnaire”, it is advised that the authors provide one or two examples of the five subscales. For example, “The skills subscale centers on strategies a patient acquires to navigate and function within the healthcare system”. It is unclear what skills are taken into consideration here. It will benefit the readers greatly if there were examples for each subscale, to better understand the factors taken into consideration.

Minor concerns :

  • Typing error (extra mid-word space) in line 20 “…Hispanic/Latino cancer patients rem ains scarce”.
  • “Relatinship BCQ’s subscales, exposure, and protection.” Please reframe the sentence.
  • Typing error in table 1, Section “Cancer Diagnosis Clinical Characteristics”: The first line says “Type of cancer breast”.
  • The word “Relationship” is misspelt at more than one places in the manuscript.

Overall, the manuscript provides a satisfactory assessment of the different kinds of barriers to accessing health care, following a natural disaster event, that may lead to more focus on improvement of these factors in a clinical setting. 

Author Response

Thank you so much for all your valuable feedback. The following observations were addressed. 

  • Reference to the parent study has been added using grant number as no publication has been done addressing the specific aims of the study.
    • Materials and Methods (line 77-78): “To test the proposed variables, the team conducted secondary analyses derived from a longitudinal case-control cohort study”. Please provide reference to the parent study, if the study has been published. 
  • Barriers to Care Questionnaire items were included as example in the description of each subscale.
    • Materials and Methods: In the section describing the “Barriers to Care Questionnaire”, it is advised that the authors provide one or two examples of the five subscales. For example, “The skills subscale centers on strategies a patient acquires to navigate and function within the healthcare system”. It is unclear what skills are taken into consideration here. It will benefit the readers greatly if there were examples for each subscale, to better understand the factors taken into consideration.

Minor concerns :

  • Typing errors were addressed 
    • Typing error (extra mid-word space) in line 20 “…Hispanic/Latino cancer patients rem ains scarce”.
    • “Relatinship BCQ’s subscales, exposure, and protection.” Please reframe the sentence.
    • Typing error in table 1, Section “Cancer Diagnosis Clinical Characteristics”: The first line says “Type of cancer breast”.
    • The word “Relationship” is misspelt at more than one places in the manuscript.

Overall, the manuscript provides a satisfactory assessment of the different kinds of barriers to accessing health care, following a natural disaster event, that may lead to more focus on improvement of these factors in a clinical setting.

Reviewer 2 Report

The issue discussed in this manuscript is very interested to most of health care providers in cancer care. However, the manuscript requires significant efforts to revise before it can be accepted.

  • Writing style: The manuscript should be carefully edited as it appears to have many typos. The introduction section should be revised to follow the general principles of academic writing with clear topic sentence and evidence to support the topic in the paragraph.
  • Abstract: The background section is longer than expected. The authors should rewrite the abstract with more clear description in methods and results. For example, the methods section should describe how the patients were recruited, and the results section should clearly show the number of patients, the scales of questionnaires before and after the intervention.
  • Method section:
  1. The authors stated that “to test the proposed variables, the team conducted secondary analyses derived from a longitudinal case-control cohort study”. However, there is no mention and citation on the parent study.
  2. How the Spanish version of Barriers to Care Questionnaire (BCQ) translated and validated should be mentioned. Please describe and cite the study performed the translation in the reference.
  • Results:
  1. The authors stated that the patients were recruited from local community oncology clinics, cancer support groups, the Puerto Rico American Cancer Society, and community-based cancer support service organizations. The eligibility criteria were clear and adequate. However, there was no exclusion criteria. As the study recruited the subjects from multiple sites, the final sample size was only 51, with majority of females. It should be clearly explained and described about the sampling method and why the distribution of gender was very uneven. Is the study sample ready to represent the population of interested? The sample size is very small and may not be able to demonstrate the general population of the study interested.
  2. Table 3 is unclear to the readers. What is 1 and 2 means in the first row of the table?
  3. The results of SPSI-D, including optimistic or positivist beliefs (creencias de suerte), positive self-concept (auto-concepto 113 positivo), disruption (disrupción), and health (salud), were not showed in this study.

Author Response

Thank you so much for all your valuable feedback. The following comments and observations were addressed in the manuscript. 

The issue discussed in this manuscript is very interested to most of health care providers in cancer care. However, the manuscript requires significant efforts to revise before it can be accepted.

  • The manuscript was revised and typos were corrected 
    • Writing style: The manuscript should be carefully edited as it appears to have many typos. The introduction section should be revised to follow the general principles of academic writing with clear topic sentence and evidence to support the topic in the paragraph.
  • The abstract was modified, shortening the introduction section, and adding more detailed to the methods section.
    • Abstract: The background section is longer than expected. The authors should rewrite the abstract with more clear description in methods and results. For example, the methods section should describe how the patients were recruited, and the results section should clearly show the number of patients, the scales of questionnaires before and after the intervention.

Method section:

  • Reference to the parent study has been added using grant number as no publication has been done addressing the specific aims of the study.
    • The authors stated that “to test the proposed variables, the team conducted secondary analyses derived from a longitudinal case-control cohort study”. However, there is no mention and citation on the parent study.
  • In the discussion section a limitation was added. The BCQ has been translated by the team as it has not been adapted for the Puerto Rican population. The alpha reliability test was added in the methods section.
    • How the Spanish version of Barriers to Care Questionnaire (BCQ) translated and validated should be mentioned. Please describe and cite the study performed the translation in the reference.

Results:

  • The sample size is small as only individuals who completed the questionaries during the seismic period were selected, as choosing participants in the only in this time range  would gather information closer to the context. As this is a brief communication the results presented do not intent to be generalized but to offer what could be preliminary data for further research on natural disasters. 
    • The authors stated that the patients were recruited from local community oncology clinics, cancer support groups, the Puerto Rico American Cancer Society, and community-based cancer support service organizations. The eligibility criteria were clear and adequate. However, there was no exclusion criteria. As the study recruited the subjects from multiple sites, the final sample size was only 51, with majority of females. It should be clearly explained and described about the sampling method and why the distribution of gender was very uneven. Is the study sample ready to represent the population of interested? The sample size is very small and may not be able to demonstrate the general population of the study interested.
  • In the result section table 3 was modified for better understanding.
    • Table 3 is unclear to the readers. What is 1 and 2 means in the first row of the table?
  • For these secondary analyses, only the exposure and the protection subscales were used, as the other subscales of the SPSI-D were not administered to the participants.
    • The results of SPSI-D, including optimistic or positivist beliefs (creencias de suerte), positive self-concept (auto-concepto 113 positivo), disruption (disrupción), and health (salud), were not showed in this study.

Reviewer 3 Report

  1. The p values indicated in the abstract are not correct and must match those reported with the data
  2. Table 2 shows the results pertaining to protection. However, description of protection is not entirely clear from the paper’s introduction, methods or discussion
  3. It might be worthwhile to include the actual survey questions as supplements, with this manuscript, to help the readers.
  4. Table 3 shows 1 and 2 as column headings, but the description of these numbers is not clear. Authors may consider including a legend along with this table.
  5. Authors may consider noting a clear distinction of this study from the parent study that investigates the effects of Hurricane Maria. What kind of information was provided to the subjects about the study and how did it differ from that of the parent study?

Author Response

Thank you so much for your valuable feedback. The following comments and observation were addressed in the manuscript. 

  • Typos were corrected 
    • The p values indicated in the abstract are not correct and must match those reported with the data
  • The protection variable was further explained in the methods and discussion section.
    • Table 2 shows the results pertaining to protection. However, description of protection is not entirely clear from the paper’s introduction, methods or discussion
  • Barriers to Care Questionnaire items were included as example in the description of each subscale.
    • It might be worthwhile to include the actual survey questions as supplements, with this manuscript, to help the readers.
  • In the result section table 3 was modified for better understanding.
    • Table 3 shows 1 and 2 as column headings, but the description of these numbers is not clear. Authors may consider including a legend along with this table.
  • Distinction between parent study and manuscript’s analyses was detailed.
    • Authors may consider noting a clear distinction of this study from the parent study that investigates the effects of Hurricane Maria. What kind of information was provided to the subjects about the study and how did it differ from that of the parent study?

Round 2

Reviewer 2 Report

The questions raised in last comments have not been corrected. Such as: 

1) 

  • Abstract: The background section is longer than expected. The authors should rewrite the abstract with more clear description in methods and results. For example, the methods section should describe how the patients were recruited, and the results section should clearly show the number of patients, the scales of questionnaires before and after the intervention.
    • Method section:
    1. The authors stated that “to test the proposed variables, the team conducted secondary analyses derived from a longitudinal case-control cohort study”. However, there is no mention and citation on the parent study.
      • Results:
      1. The authors stated that the patients were recruited from local community oncology clinics, cancer support groups, the Puerto Rico American Cancer Society, and community-based cancer support service organizations. The eligibility criteria were clear and adequate. However, there was no exclusion criteria. As the study recruited the subjects from multiple sites, the final sample size was only 51, with majority of females. It should be clearly explained and described about the sampling method and why the distribution of gender was very uneven. Is the study sample ready to represent the population of interested? The sample size is very small and may not be able to demonstrate the general population of the study interested.
      2. Table 3 is unclear to the readers. What is 1 and 2 means in the first row of the table?

Author Response

Thank you once again for your valuable feedback. I hope the changes made adequately address the suggestions offered. Bellow a point by point answer to the concerns. 

  • Reference to the parent study has been highlighted (grant number). It is important to note that no publication has yet been done addressing the specific aims of the parent study (manuscript in progress), therefore there is no citation.
  • The abstract was modified, shortening the background section, and further detailing the methods section.
  • More detail was added to describe the eligibility criteria.
  • The sample size is small as the data pooled corresponded only to individuals who completed the questionaries during the seismic period, as it would gather information closer to the context. As this is a brief communication the results presented do not intend to be generalized but to offer what could be preliminary data for further research on natural disasters.
  • In the result section in table 3, the variables were added for better understanding.
  • A clearer distinction between the parent study and the manuscript’s analysis in the methods section was detailed.
  • The manuscript was revised, and some redaction style modifications were made.